# Integrated Biobanking and Tumor Model Establishment of Human Colorectal Carcinoma Provides Excellent Tools for Preclinical Research

**DOI:** 10.3390/cancers11101520

**Published:** 2019-10-09

**Authors:** Christina S. Mullins, Bianca Micheel, Stephanie Matschos, Matthias Leuchter, Florian Bürtin, Mathias Krohn, Maja Hühns, Ernst Klar, Friedrich Prall, Michael Linnebacher

**Affiliations:** 1Molecular Oncology and Immunotherapy, Department of General, Visceral, Vascular and Transplantation Surgery, University Medical Center Rostock, Schillingallee 69, 18057 Rostock, Germany; christina.mullins@med.uni-rostock.de (C.S.M.); bianca.micheel@med.uni-rostock.de (B.M.); stephanie.matschos@med.uni-rostock.de (S.M.); mathias.krohn@med.uni-rostock.de (M.K.); 2Department of General, Visceral, Vascular and Transplantation Surgery, University Medical Center Rostock, University of Rostock, Schillingallee 35, 18057 Rostock, Germany; matthias.leuchter@med.uni-rostock.de (M.L.); florian.buertin@med.uni-rostock.de (F.B.); ernst.klar@med.uni-rostock.de (E.K.); 3Institute of Pathology, University Medical Center Rostock, Strempelstraße 10, 18057 Rostock, Germany; maja.huehns@med.uni-rostock.de (M.H.); friedrich.prall@med.uni-rostock.de (F.P.)

**Keywords:** biobanking, permanent cell line, colorectal carcinoma, patient derived xenograft, primary cell culture, tumor model, precision medicine

## Abstract

Over the time period from 2006 to 2017, consecutive patients operated on at the University Medical Center Rostock participated in the comprehensive biobanking and tumor-modelling approach known as the HROC collection. Samples were collected using strict standard operating procedures including blood (serum and lymphocytes), tumor tissue (vital and snap frozen), and adjacent normal epithelium. Patient and tumor data including classification, molecular type, clinical outcome, and results of the model establishment are the essential pillars. Overall, 149 patient-derived xenografts with 34 primary and 35 secondary cell lines were successfully established and encompass all colorectal carcinoma anatomic sites, grading and staging types, and molecular classes. The HROC collection represents one of the largest model assortments from consecutive clinical colorectal carcinoma (CRC) cases worldwide. Statistical analysis identified a variety of clinicopathological and molecular factors associated with model success in univariate analysis. Several of them not identified before include localization, mutational status of K-Ras and B-Raf, MSI-status, and grading and staging parameters. In a multivariate analysis model, success solely correlated positively with the nodal status N1 and mutations in the genes K-Ras and B-Raf. These results imply that generating CRC tumor models on the individual patient level is worth considering especially for advanced tumor cases with a dismal prognosis.

## 1. Introduction

Establishment of tumor models has a long-standing history. The first patient derived tumor cell line, i.e., HeLa, was established in 1951 [1]. The patient (Henrietta Lacks), a 30-year old female with cervical carcinoma, and her cell line are famous around the world. Since then, “uncountable” in vitro models have been established. In vivo tumor models followed suit in the 1960s. In this case, patient individual models, the so-called patient derived xenografts (PDX), are a recent trend [2].

The preclinical screening method of choice was, for a very long time, the NCI-60 cell line panel. It consists of a diverse group of cancer cell lines established by the National Cancer Institute for the purpose of anticancer drug discovery. Since 1990, more than 100,000 compounds have been screened [3]. Still, not more than 8% of first-in-human compounds actually reach the step of FDA registration [4]. Reasons for this dismal rate are especially: (I) “one drug does not fit all” or the negligence of properly identifying responders [4], (II) using cell lines of unknown but high passages, which tolerates the risk of culture artifacts such as relevant changes in biological behavior and molecular composition and (III) misidentification of cell lines (epithelial cell lines are often actually HeLa cells and lymphoid cell lines are often actually Jurkat cells) [5]. 

To overcome the desolate drug-development results and frequent failure in the transfer of preclinical data into successful clinical trials, PDX patients in the form of 1 × 1 × 1 experimental design (PDX clinical trial (PCT)) have gained great popularity. By implementing the ’one animal per model per treatment’ approach inter-patient response heterogeneity could be depicted accurately. Furthermore, Gao and colleagues could demonstrate both the reproducibility and clinical translatability by identifying associations between genotypes and drug responses, and established mechanisms of resistance [6]. 

In parallel, the generation of large biobanks, which enable individualized therapy approaches, at least on a patho-molecular level of the tumor, has become standard in comprehensive cancer centers. Bedra and colleagues demonstrated that the integration of cancer registry and clinical research departments is beneficial, especially since there is increased knowledge, manpower, and crossover in job responsibilities [7]. 

Lastly, critical factors and pieces of information for successful implementation of tumor models in a drug response prediction are: the models’ number of passages, culture condition (2D vs. 3D vs. organoids, subcutaneously vs. orthotopic), molecular composition, and subtype [8].

We report on our experience of more than 10 years of combined biobanking and tumor modelling for colorectal carcinomas (CRC). Special emphasis is put on standard operating procedures (SOP) establishment for optimal biobanking and technical refinement for a tumor model establishment (both in vitro and in vivo).

## 2. Results

### 2.1. Patient Cohort Characteristics

Over the time period from 2006 to 2017, consecutive patients diagnosed with CRC operated on at the university medicine of Rostock participated in this comprehensive biobanking and tumor-modelling approach: the HROC collection (HRO: Hansestadt Rostock, C: colorectal cancer). Samples were collected using strict SOP including blood (serum und lymphocytes), tumor tissue (vital and snap frozen), and adjacent normal epithelium [9]. Patient and tumor data collected include classification, clinical course and outcome, molecular subtype, and results of model establishment (See Table 1 and Appendix A). Overall, the patient population consisted of 382 patients diagnosed with a colorectal neoplasia. Tissue collection was attempted for adenomas as well as primary and metastatic CRC. For several patients, more than one sampling attempt was undertaken. The exact details are given in Appendix A. The CRC patient population (n = 350, patients with only adenoma excluded) had a median age of 72 years, ranging from 21 to 98 years, and consisting of 59.1% male and 40.9% female. The incidence of stage I-IV CRC according to the Union Internationale Contre le Cancer (UICC) classification was 21.1%, 32.0%, 24.9%, and 21.7%, respectively. A single patient showed complete remission after neoadjuvant chemoradiation and was classified as UICC 0. The successfully bio-banked material comprises 262 primary resected colorectal adenocarcinomas as well as 44 metastases from CRC. In addition, nine adenomas were collected during this period. Metastatic CRC tissue originated from the liver (n = 36, 81.8%), peritoneum (n = 4, 9.1%), brain (n = 2, 4.5%), lung (n = 1, 2.3%), and non-regionary lymph nodes (n = 1, 2.3%). Our study population was consistent with the general CRC population of Western European countries with regard to age, gender, and stage distribution [10]. However, compared to epidemiological data concerning the relative frequency of primary colorectal tumor sites, our specimen collection overrepresents tumors of the proximal colon (46.9%) and underrepresents rectal tumors (13.7%) [11,12]. Successfully harvested CRC tissue originated from the rectum (13.7%), the sigmoid (22.5%), the left colon (13.0%), the transverse colon (5.3%), and the right colon (45.4%). The relative paucity of rectal cancers is due to the frequent inability to yield sufficient amounts of tumor tissue after neoadjuvant treatment. From the total of 350 included CRC patients’ 5-year follow-up data was available for 236 cases (67.4%). The 5-year survival rates were 72.3% for stage I, 66.2% for stage II, 53.8% for stage III, and 9.6% for stage IV disease. The stage adjusted survival rates of our patient set are considerably poorer compared to the survival rates of other Western European CRC-patients’ cohorts [13]. This can be attributed, to relatively low follow-up rates, which emphasizes cases with metastatic disease recurrence and to the sampling restricted to a resection specimen. Thus often no tumor material from smaller tumors is obtained. To complete the information, the unsuccessful attempts to collect tumor material were also listed in Table 1.

### 2.2. Establishment of Patient-Derived Cell Lines (Primary Cell Lines)

Establishment of a primary (patient derived) CRC cell line was attempted for 294 tumors. In 94 additional cases, no cell suspension could be prepared either due to tissue quality (i.e., necrosis) as assessed by a trained pathologist or due to tissue shortage. Unfortunately, microbial contamination (generally due to “contaminated” tumor tissue) occurred in 31 primary cell suspensions. The overall success rate for permanent cell lines was 34/263 (12.9%, contaminated primary cultures were not counted). After seeding of single cell suspensions on collagen-coated plates, no signs of viable cells could be observed for about one-third of cell suspensions. In the remaining two-thirds of cases, cells initially attached to the culture dishes. However, nearly half of the plates (≅ 1/3 of total attempts) had to be disposed of after about one month due to a lack of cell proliferation and massive cell death. Culture plates to which cells had attached initially and for which cell proliferation could be observed (≅ 1/3 of total attempts) were maintained under close surveillance. Growth of tumor colonies ranged from very rapid (days) to extremely slow (several months). More permanent patient-derived cell lines were established from slow proliferating cells. In total, this procedure led to the generation of 34 permanent patient-derived cell lines (Figure 1 and Appendix A).

We, thus, investigated if any clinical, molecular, or other parameters could be identified that correlate with the success rate and found that solely higher tumor grading G (*p* = 0.007) was positively associated with a primary cell line establishment success (Table 2A). A trend for higher success in a model establishment was observed for the advanced nodal status N (*p* = 0.057). No influence was observed for patients’ gender, age, or sample type (carcinoma or metastasis) at the time of surgery. Neither did tumor localization (right, transverse, left, sigma, or rectum), molecular subtype (microsatellite status, CpG island methylation, or chromosomal instability), MSI status, staging parameters: T, M, R, L, and V, UICC classification nor genetic mutations in TP53, APC, K-Ras, B-Raf, and PIK3CA seem to be relevant.

### 2.3. Establishment of Patient-Derived Xenografts (PDX)

In parallel to a primary cell line establishment, engraftment of tumor tissue into immunodeficient mice, so called PDX, was performed. In total, engraftments were attempted from 219 samples. Successful tumor outgrowth occurred in 149/219 (68.0%) cases.

Statistical analyses of these 219 cases revealed that patient age (*p* = 0.048), sample type: metastatic sample versus primary colon carcinoma (*p* = 0.014), gene mutations in K-Ras (*p* = 0.019) and B-Raf (*p* = 0.002), and the staging parameter M (*p* = 0.047) were positively correlated with PDX success. A clear correlation was also observed with tumor localization (*p* < 0.001), molecular subtype (*p* = 0.003), and a microsatellite instable (MSI) status (*p* = 0.001) (Table 2B). A trend was observed for the tumor grading G (*p* = 0.083), and the nodal status N (*p* = 0.062) as well as the resection extent R (*p* = 0.074). No significances were observed for staging parameters: T, L, and V as well as mutations in the genes TP53, APC, and PIK3CA. 

### 2.4. Establishment of PDX-Derived Cell Lines (Secondary Cell Lines)

After successful PDX generation (in parallel to further PDX expansion for biobanking), secondary (=PDX-derived) cell lines were established. All but one of the PDX cases (n = 148) were included and led to the generation of 35 secondary cell lines (Figure 1). Thus, the success rate almost doubled compared to the primary cell lines with 34/263 (12.9%) versus 35/148 (23.6%) for primary and secondary cell lines, respectively. Both primary and secondary cell line could be established for 12 tumor samples. An additional 23 (secondary) cell lines complete the HROC patient-derived cell line collection (in total, CRC cell lines from 57 different tumor samples included in this study). These “additional” cell lines include one case for which no primary cell line could be established due to microbial contamination and four cases for which insufficient tumor material was available for biobanking. Since the success rate(s) of PDX (as well as secondary cell line) generation surpasses that of primary cell lines (68.0% (and 23.6%) vs. 12.9%), PDX generation is given priority in such cases.

In this fairly small sampling, only the mutation status of the genes APC (*p* = 0.005) and B-Raf (*p* = 0.009) correlated with establishment success of secondary cell lines. All other parameters did not significantly influence the success of a secondary cell line establishment (data not shown). 

### 2.5. Global Patient and Model Analysis

In total, we could establish at least one patient-individual model (cell line and/or PDX) for 153/306 (50.0%) tumor samples. For four samples, only a primary cell line, for 96, only PDX and, for 53 samples, the cell line (primary or secondary) and PDX could be established. Of note, in 12 cases, we succeeded to establish all three model types (PDX as well as primary and secondary cell lines).

Overall success (of at least one patient-individual model successfully established) was significantly influenced by tumor localization (*p* = 0.011), staging parameters N (*p* = 0.011), M (*p* = 0.008), G (*p* = 0.006), and R (*p* = 0.026), the UICC stage (*p* = 0.023), mutation status of the genes K-Ras (*p* = 0.014) and B-Raf (*p* = 0.022), the molecular subtype (*p* = 0.024), and the MSI status (*p* = 0.019) (Table 2C). A trend for the staging parameter T (*p* = 0.061) and tumor model establishment was observed.

Of note, in a multivariate analysis, solely the nodal status N (primary cell lines: *p* = 0.026, PDX: *p* = 0.024, all models combined: *p* = 0.046) and mutation status of the genes K-Ras (PDX: *p* = 0.005, all models combined: *p* = 0.002) and B-Raf (PDX: *p* = 0.004, all models combined: *p* = 0.026) were significant when compared to multiple statistically significant correlations observed in a univariate analysis.

The model establishment neither correlated with patients´ overall survival (OS) nor relapse-free survival (RFS) (Figure 2).

Albeit not the focus of the current manuscript, it is worth mentioning that a model establishment was attempted for subsidiary neuroendocrine tumor samples (three primaries and two metastases) listed in Appendix A but not included in the statistical analysis of the CRC model success. One cell line and matching PDX could be stablished [14].

## 3. Discussion

An average of about 10% establishment rate for patient-derived CRC cell lines growing in 2D from primary tissue is expected for larger cohort studies [15]. In the present series, we observed a slight increase in the success rate over time, but this was not significant [data not shown]. The majority of the cell line generation attempts have been performed without the addition of ROCK inhibitors and no feeder cells were used at all, despite the fact that, while performing the study, a very successful 2D culture system was suggested by Liu and colleagues [16]. Secondary cell line generation attempts using PDX tissue as starting material allowed for almost a duplication of the success rate (23.6% versus 12.9%), which is still lower than the nearly 50% reported by a French group [15]. This might be attributable to differences in the cell line establishment protocols, but it might also reflect the fact that we tried to establish secondary cell lines from 148 PDX cases obtained, including many from tumors in lower stages.

Whereas the higher success rate of a secondary stage in comparison to primary cell line generation can be attributed to (I) more material available for culturing (II) multiple attempts, it is possible to use several PDX cases from the same tumor, and (III) potentially because more vital and actively growing tumor cells are typically present in PDX when directly compared to the original tumor tissues. Compiled, this led to the generation of 57 individual cell lines and 12 sets of cell lines established from primary tissue as well as from PDX. These pairs might be useful to prove if there is no relevant difference between primary and secondary patient-derived cell lines, as suggested by Dangles-Marie and coworkers [15].

Of note, the HROC model establishment is an ongoing activity and we regularly report new permanent low-passage HROC cell lines with accompanying data to Cellosaurus [17].

Our success rate in PDX generation improved slightly over time from about 40% measured on the individual immunodeficient mouse level in T0 to 48% at present [March 2019, data not shown]. This is similar to the observed trend in generating HROC cell lines and, taken together, this very likely reflects a flat but perceivable learning curve. The latter implies that the complete data set is not heavily influenced by experimental bias due to different experimenters and experimental conditions. 

The aggregated overall success rate of HROC PDX generation was 68.0% (149/219 cases), which is definitely superior to the cell culture for CRC. If patient-tumor-material is not “limited” and, thus, repeated implantations can be performed in severely immunodeficient mice, even higher success rates are achievable. However, the 60+% are a realistic figure for large-scale PDX establishment series and correspond to or surpass similar data sets in literature [18,19,20,21]. Higher [22,23] but also lower rates e.g., [24] have been described for engraftment of tumor pieces as well as “secondary” xenograft models established from patient-derived organoids (PDO).

One of the limitations of the data presented lies in the fact that we did not perform such PDO-generation partly because this technique emerged while the study was ongoing and partly because we decided to focus on the more classical and widely used patient-derived CRC models. This includes 2D cell lines for in vitro and PDX for in vivo analysis. However, due to the fact that, from n = 262, individual tumors plus n = 44 metastases vitally frozen samples are stored in the HROC biobank, PDO generation could still be attempted. In addition, vitally frozen normal tissue is available from n = 81 patients, totaling n = 70 primary tumor and normal tissue pairs and n = 15 metastases and normal tissue pairs. These cases, as well as our general strategy of biobanking vital tissues from cancer patients, might be of interest when searching for rare genotypes to generate PDO in order to establish larger organoid collections for drug response screens, as recently suggested [25]. Similarly, using these vitally cryopreserved tumor pieces, it would be possible to establish cultures from further tumor-associated cell types present in the tumor micromilieu including endothelial cells, tumor-associated fibroblasts, and macrophages as well as tumor-infiltrating B and T cells. 

An additional restraint arises from the very limited functional data included in the present manuscript. Several of the HROC cell lines as well as PDX models have already been published [14,26,27,28,29]. For the PDX collection, a subset of >100 cases is currently under detailed analysis and this study will be published soon. Furthermore, a first PCT study is currently ongoing. However, several of the HROC models have been used successfully in published preclinical research studies [30,31]. 

One major goal of our study was to establish tumor models from all histological sides, stages, and possibly, most importantly, known subtypes of CRC since this is a central requirement for precision medicine. From the molecular point of view, this goal was clearly achieved, since chromosomal instability (CIN), CpG island methylator phenotype (CIMP), MSI-L (low), and MSI-H (high) cases, both sporadic and with a Lynch syndrome (LS) background, could be successfully established. Similarly, the models reflect the large variety in mutations in different driver genes of CRC like K-Ras, B-Raf, TP53, PIK3CA, and APC [32]. In contrast, we failed to establish models from the two familial adenomatous polyposis (FAP) tumors included in the study. These tumors were obtained synchronous from one patient (HROC372) and due to both a low amount and a low quality of the tissues, there was only one unsuccessful primary cell line establishment attempt. 

A major focus of the present study was to perform a comprehensive correlation analysis of modeling success with clinical and molecular data available for the HROC collection. Only a few studies analyzed such correlations for CRC before and all of them looked for factors influencing PDX success. Julien et al. established PDX from 35/58 samples (60.3%), but described significant correlations of PDX success only for the combination of N1/N2 + CEA>6 ng/ml + proximal or distal localization [18]. Thus, a direct comparison with our results is not possible. Another small study was performed by a French group (35/40 CRC established as PDX, 87.5%) and they found a significant correlation of PDX success with advanced N and G status (*p* = 0.017 and 0.0206, respectively) [33]. A very recent US study focused on finding significant factors influencing PDX success in CRC [20]. With 50 out of 90 tumor cases successfully established as PDX (55.6%), it is a medium size study. However, it is clearly biased by a very high proportion of metastases included (75/90, 83.3%). One significant correlation observed was the resection specimen versus the biopsy (*p* = 0.001) and, since we did not include biopsies in our study, this cannot be compared directly to our data. Moreover, albeit only significant in univariate analysis, this study found a significant correlation for a primary tumor versus metastasis (*p* = 0.010). Furthermore, the success was higher for primary tumor samples (13/15, 86.7% versus 37/75, 49.3%), whereas we found exactly the opposite to be significant (*p* = 0.014). The only reasonable explanation for this difference might be a very high proportion of biopsies for the metastatic tissues in the US study. Yet, our data clearly suggest that, when using tissue of a high quality from surgical resections, the PDX success rate of metastases is higher than that of primary tumors. 

Another study from Korea generated 150 PDX from 241 patients (62.2%) and observed significant correlations with tumor stage (*p* < 0.001) and grading (*p* = 0.029) [19]. Both findings are divergent from ours, since our PDX success was not significantly correlated with T, G, R, L, and V. However, for N, only a trend was observed (*p* = 0.062). Our significant findings also analyzed factors that were found to be insignificant in the Korean study. These factors were patient age (*p* = 0.048), tumor localization (*p* < 0.001), MSI status (*p* = 0.001), and metastatic status M (*p* = 0.047). Since the study sizes as well as PDX success rates are almost equal, one might speculate that these discrepancies are attributable to the different racial background and maybe partly due to the fact that they did not include metastases into their analysis. In their correlation analysis of PDX success and clinical outcome, measured as cumulative survival after three years, they found a positive correlation of PDX success with a worse clinical outcome [19]. This finding prompted us to similarly analyze the correlation of survival and clinical outcome for PDX and an additionally primary cell line as well as “any” model success with a follow-up time of at least five years in our cohort. However, no significant correlation between model success (primary cell line, PDX or “any” model) and clinical outcome was observed after three years (*p* = 0.83 for UICC stage III and *p* = 0.73 for UICC stage IV and PDX success) nor after five or more years of follow-up. This might be attributable to the fact that there was a bias toward larger and, thus, frequently more advanced tumors, which were successfully collected, plus the relatively high rate of model success of the lower stage tumor samples. 

When aggregating and comparing the findings of these previous studies with the results of the present analysis, we would like to conclude that we were, for the first time, able to identify several clinical parameters, which are likely to influence the success rate of PDX establishment from CRC resection specimens. These parameters were patient age (*p* = 0.048), tumor localization (*p* < 0.001), higher M status (*p* = 0.047), molecular subtype (*p* = 0.003), especially the MSI status (*p* = 0.001), mutations in K-Ras (*p* = 0.019), and B-Raf (*p* = 0.002) as well as a significantly better engraftment of metastatic (*p* = 0.014) tissues. 

A significant positive correlation was additionally noticed for high tumor grading G (*p* = 0.007) with the successful establishment of a patient-derived 2D tumor cell line. 

It is of interest that, when combining the success data from PDX and primary cell line establishment, several factors lost their significance: patient age and sample type (primary versus metastatic tissue). Besides mutated TP53 (*p* = 0.019), factors associated with a more advanced tumor stage: higher N, M, and R as well as the UICC stage reached the level of positive correlation with overall model success (*p*-values of 0.011, 0.008, 0.026, and 0.023, respectively) in addition to higher grading (*p* = 0.006 for any model success), which was significant for the primary cell line but not PDX success. This finding led us to hypothesize that even higher sample numbers are necessary to identify all factors significantly influencing CRC model success.

Additionally, the nodal status (N1) and mutations in the genes K-Ras and B-Raf of the factors influencing model establishment success in univariate analysis endured when using multivariate analysis (*p* = 0.024, *p* = 0.002, and *p* = 0.026, respectively). This is completely in line with all similar previous studies and might be best explained by the low numbers of cases analyzed. However, when interpreting more optimistically, this fact combined with the overall high success rate of the CRC model establishment implies that generation of tumor models on an individual patient level – one pillar of modern precision medicine – is not exceedingly biased by clinicopathological and molecular factors and, consequently, could and should be attempted for patients with advanced disease and/or a dismal prognosis. 

## 4. Materials and Methods

Surgically resected primary CRC tissue, corresponding metastatic tissue, and samples from liver, lung, brain, peritoneal, and lymph node metastasis, as well as adenomas, were obtained from consenting patients at the university hospital of Rostock from 2006 to 2017. The study was approved by the ethics committee of the Rostock University Medical Center (II HV 43/2004 and A 45/2007) [9].

In vitro cultures from (I) CRC patients’ tumors or (II) PDX tumors were prepared as previously described [28]. The protocol evolved over time but, roughly, the procedure was as follows: tumor tissue was minced mechanically by crossed scalpels in PBS, cell suspension was passed through a cell strainer (100 µm Nylon mesh), washed, resuspended in tumor cell culture medium (Quantum 263 (PAA, Pasching, Austria) or MOITUM (DMEM/Ham’s F12 supplemented with non-essential aminoacids, insulin-transferrin-sodium selenite and BME vitamins) supplemented with 2 mM L-glutamine, antibiotics, and antimycotics (penicillin, streptomycin, gentamicin, and amphotericin B), and seeded on collagen-coated six-well plates.

All animal experimental procedures were approved by the National Animal Welfare Committee (permit numbers: LALLF M-V/TSD/7221.3-1.1-071/10, 7221.3-1-015/14, 7221.3-2-020/17). PDX from CRC patients´ tumors were generated, as described previously [34]. Additionally, 3x3x3 mm cubes of tumor tissue were implanted into the flanks of immunodeficient mice. Tumor tissue was implanted either (I) directly after a patient’s surgery or after (II) cryo-conservation. The following mouse strains used in this study: (I) NOD/SCID, (II) NMRI-Foxn1nu (NMRI), and (III) NOD.Cg-Prkdcscid Il2rgtm1Wjl/SzJ (NSG).

Molecular classification included: Microsatellite instability and methylation status of CpG islands in promoter regions, which were performed as described previously [35]. The mutational status of KRAS exon 2–4, BRAF exon 15, TP53 exon 5–8, APC exon 16, and PIK3CA exon 10 and 21 were tested, as previously described [36].

Statistical analysis was performed either using the software package “R” version 3.5.2 (for correlations with RFS and OS) [37] or the statistical program IBM SPSS Statistics 25 (correlation of model establishment with clinicopathological parameters). Survival rates were analyzed using the Kaplan–Meier method and log-rank test. Multivariate analysis was performed using logistic regression to identify predictors for the success rate of the patient-derived tumor model and cox-regression to identify determinants for RFS and OS. For comparability, we used standardized definitions of end points for RFS [38]. For the correlation analysis, both the chi-squared-test and the Fisher exact test were applied.

## 5. Conclusions

In sum, this study succeeded in generating CRC in vitro and in vivo models from all subtypes of CRC with the exception of FAP. To the best of our knowledge, the HROC collection represents the largest single-center integrated biobanking activity of CRC-patient biomaterial along with individual-patient-derived in vitro and matching in vivo models worldwide. Several clinical and molecular factors significantly influencing the success of model generation were identified with many for the first time. All models are available upon reasonable request.

## Figures and Tables

**Figure 1 cancers-11-01520-f001:**
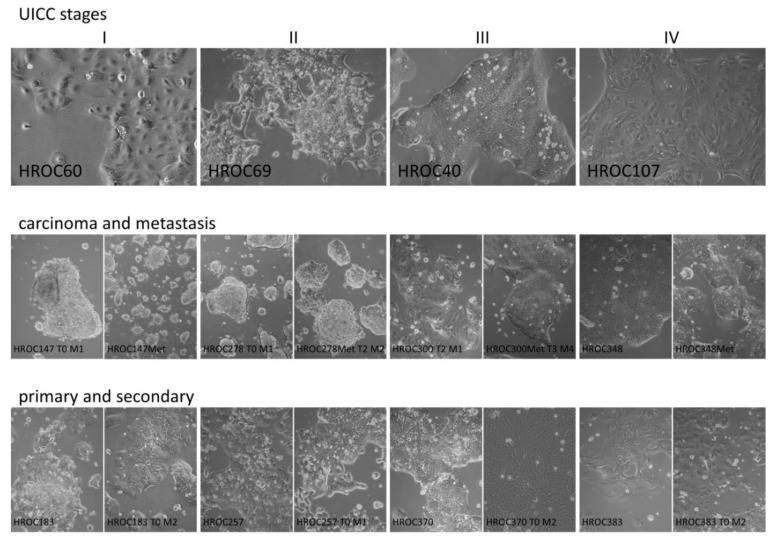
The figure depicts exemplarily the morphology of cell lines derived from tumors with UICC stages I, II, III, and IV (top, left to right), cell line pairs derived from carcinoma and corresponding metastasis (middle), and pairs of primary and corresponding secondary cell lines (bottom). The images were taken with a 10× lens of a Primo Vert microscope (Carl Zeiss Microscopy GmbH, Jena, Germany).

**Figure 2 cancers-11-01520-f002:**
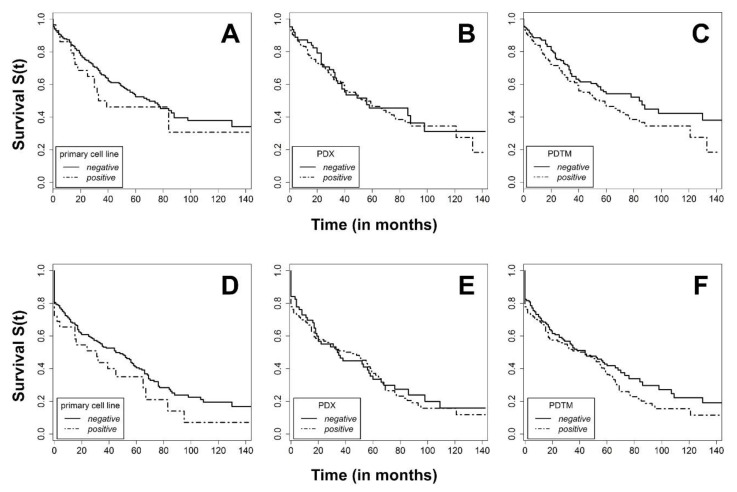
Kaplan–Meier curve for overall survival (OS) (top) of patients for whom (**A**) a primary cell line, (**B**) a PDX, or (**C**) any type of model could be established from their respective tumor tissue and relapse-free survival (RFS, bottom) of patients for whom (**D**) a primary cell line, (**E**) a PDX or (**F**) any type of model (patient-derived tumor model, PDTM) could be established from their respective tumor tissue (positive line) or not (negative line).

**Table 1 cancers-11-01520-t001:** Biobanking and patient overview.

Patients Characteristics (n = 382)						
	Male	Female	Mean Age	5-year Follow-up Rate	5-year Relapse-free Survival	5-year Survival Rate
Adenoma (n = 32)	23 (71.9%)	9 (28.1%)	69.0 (28–85)	n.a.	n.a.	n.a.
CRC (n = 350 ^◆^)	207 (59.1%)	143 (40.9%)	72.0 (21–98)	236 (67.4%)	43.2%	51.3%
UICC I (n = 74; 21.1%)	42 (56.8%)	32 (43.2%)	73.0 (28–98)	47 (63.5%)	61.7%	72.3%
UICCII (n = 112; 32.0%)	63 (56.2%)	49 (43.8%)	73.5 (21–92)	71 (63.4%)	62%	66.2%
UICC III (n = 87; 24.9%)	47 (54.0%)	40 (46.0%)	70.0 (40–88)	65 (74.7%)	43%	53.8%
UICC IV (n = 76; 21.7%)	54 (71.0%)	22 (29.0%)	70.0 (30–85)	52 (68.4%)	0%	9.6%
**Sample Properties (n = 315)**						
		**primary resected tumors**			**metastases**	
Adenoma (n = 9)		9			/	
Adenocarcinoma (n = 306)		262			44	
**Unsuccessful tissue acquisition (n = 107)**						
		**primary resected tumors**			**metastases**	
Adenoma (n = 25)		25			/	
Adenocarcinoma (n = 82)		72			10	
**CRC Metastases Characteristics (n = 44)**						
	**resection site**					
	liver	lung		peritoneum	brain	lymph node
synchronous (n = 18)	17	0		1	0	0
metachronous (n = 26)	19	1		3	2	1
initial stage	UICC I	UICC II		UICC III	UICC IV	
	5	4		10	7	
sets of primary tumor and one or more corresponding metastases			N = 20			

The biobank includes samples from 382 patients, who consented to donate one or more tissue sample. One patient presented with a secondary tumor during follow-up and was classified as two individual cases (HROC50). ^◆^ One Patient showed complete remission of a rectal carcinoma and was classified as UICC 0. Tissue acquisition failed due to small neoplasia size in 107 (25 adenomas, 72 carcinomas, and 10 metastases) cases.

**Table 2 cancers-11-01520-t002:** Statistical correlation analyses.

A: Primary Cell Line					
	patients	model success		*p*-value	
n (%)		no	yes	univariate	multivariate
**age**				0.715	
<61	54 (20.5)	45 (83.3)	9 (16.7)		
61–70	65 (24.7)	58 (89.2)	7 (10.8)		
71–80	105 (39.9)	93 (88.6)	12 (11.4)		
>80	39 (14.8)	33 (84.6)	6 (15.4)		
**gender**				0.559	
male	159 (60.5)	140 (88.1)	19 (11.9)		
female	104 (39.5)	89 (85.6)	15 (14.4)		
**sample type**				0.197	
primary	221 (84.0)	195 (88.2)	26 (11.8)		
metastasis	42 (16.0)	34 (81.0)	8 (19.0)		
**localization**				0.156	
left colon	26 (9.9)	23 (88.5)	3 (11.5)		
rectum	29 (11.0)	28 (96.6)	1 (3.4)		
right colon	101 (38.4)	84 (83.2)	17 (16.8)		
sigmoid	54 (20.5)	51 (94.4)	3 (5.6)		
transverse	11 (4.2)	9 (81.8)	2 (18.2)		
metastasis	42 (16.0)	34 (81.0)	8 (19.0)		
**T**				0.778	
T0-2	38 (17.4)	34 (89.5)	4 (10.5)		
T3-4	181 (82.6)	159 (87.8)	22 (12.2)		
**N**				0.057	**0.026 ***
N0	113 (51.6)	101 (89.4)	12 (10.6)		
N1	44 (20.1)	42 (95.5)	2 (4.5)		
N2	62 (28.3)	50 (80.6)	12 (19.4)		
**M**				0.775	
M0	165 (75.3)	146 (88.5)	19 (11.5)		
M1	54 (24.7)	47 (87.0)	7 (13.0)		
**G**				0.007 **	
G1-2	151 (68.9)	139 (92.1)	12 (7.9)		
G3-4	68 (31.1)	54 (79.4)	14 (20.6)		
**R**				0.769	
R0	172 (78.5)	153 (89.0)	19 (11.0)		
R1	7 (3.2)	6 (85.7)	1 (14.3)		
R2	40 (18.3)	34 (85.0)	6 (15.0)		
**L**				0.657	
L0	155 (71.8)	139 (88.0)	17 (11.0)		
L1	61 (28.2)	53 (86.9)	8 (13.1)		
**V**				0.769	
V0	128 (59.3)	112 (87.5)	16 (12.5)		
V1	82 (38.0)	74 (90.2)	8 (9.8)		
V2	6 (2.7)	5 (83.3)	1 (16.7)		
**molecular type**				0.707	
spStd	84 (56.4)	74 (88.1)	10 (11.9)		
spMSI-H	25 (16.8)	20 (80.0)	5 (20.0)		
CIMP-H	30 (20.1)	25 (83.3)	5 (16.7)		
Lynch	10 (6.7)	8 (80.0)	2 (20.0)		
**UICC**				0.896	
I	35 (14.7)	31 (88.6)	4 (11.4)		
II	79 (33.2)	71 (89.9)	8 (10.1)		
III	65 (27.3)	56 (86.2)	9 (13.8)		
IV	59 (24.8)	51 (86.4)	8 (13.6)		
**MSI status**				0.297	
MSS + MSI-L	109 (75.7)	95 (87.2)	14 (12.8)		
MSI-H	35 (24.3)	28 (80.0)	7 (20.0)		
**TP53**				0.121	**0.105**
wt	15 (40.5)	13 (86.7)	2 (13.3)		
mut	22 (59.5)	14 (63.6)	8 (36.4)		
**K-Ras**				0.248	
wt	106 (65.4)	94 (88.7)	12 (11.3)		
mut	56 (34.6)	46 (82.1)	10 (17.9)		
**B-Raf**				0.255	
wt	139 (85.3)	122 (87.8)	17 (12.2)		
mut	24 (14.7)	19 (79.2)	5 (20.8)		
**PIK3CA**					
wt	27 (100.0)	23 (85.2)	4 (14.8)		
mut	0 (0.0)	0 (0.0)	0 (0.0)		
**B: PDX**					
	**patients**	**model success**		***p*-value**	
**n (%)**		**no**	**yes**	**univariate**	**multivariate**
**age**				0.048 *	
<61	49 (22.4)	22 (44.9)	27 (55.1)		
61–70	51 (23.3)	16 (31.4)	35 (68.6)		
71–80	78 (35.6)	17 (21.8)	61 (78.2)		
>80	41 (18.7)	15 (36.6)	26 (63.4)		
**gender**				0.196	
male	127 (58.0)	45 (35.4)	82 (64.6)		
female	92 (42.0)	25 (27.2)	67 (72.8)		
**sample type**				0.014 *	**0.092**
primary	188 (85.8)	66 (35.1)	122 (64.9)		
metastasis	31 (14.2)	4 (12.9)	27 (87.1)		
**localization**				0.000 ***	
left colon	19 (8.7)	4 (21.1)	15 (78.9)		
rectum	32 (14.6)	17 (53.1)	15 (46.9)		
right colon	82 (37.4)	20 (24.4)	62 (75.6)		
sigmoid	43 (19.6)	24 (55.8)	19 (44.2)		
transverse	12 (5.5)	1 (8.3)	11 (91.7)		
metastasis	31 (14.2)	4 (12.9)	27 (87.1)		
**T**				0.293	
T0-2	27 (14.5)	12 (44.4)	15 (55.6)		
T3-4	159 (85.5)	54 (34.0)	105 (66.0)		
**N**				0.062	
N0	94 (50.5)	41 (43.6)	53 (56.4)		
N1	39 (21.0)	10 (25.6)	29 (74.4)		0.024 *
N2	53 (28.5)	15 (28.3)	38 (71.7)		0.200
**M**				0.047 *	**0.035 ***
M0	136 (73.1)	54 (39.7)	82 (60.3)		
M1	50 (26.9)	12 (24.0)	38 (76.0)		
**G**				0.083	**0.110**
G1-2	123 (66.1)	49 (39.8)	74 (60.2)		
G3-4	63 (33.9)	17 (27.0)	46 (73.0)		
**R**				0.074	
R0	140 (75.3)	56 (40.0)	84 (60.0)		
R1	7 (3.8)	2 (28.6)	5 (71.4)		
R2	39 (21.0)	8 (20.5)	31 (79.5)		
**L**				0.996	
L0	133 (72.3)	47 (35.3)	86 (64.7)		
L1	51 (27.7)	18 (35.3)	33 (64.7)		
**V**				0.506	
V0	108 (58.7)	35 (32.4)	73 (67.6)		
V1	72 (39.1)	29 (40.3)	43 (59.7)		
V2	4 (2.2)	1 (25.0)	3 (75.0)		
**molecular type**				0.003 **	
spStd	82 (53.9)	34 (41.5)	48 (58.5)		
spMSI-H	27 (17.8)	3 (11.1)	24 (88.9)		
CIMP-H	34 (22.4)	9 (26.5)	25 (73.5)		
Lynch	9 (5.9)	0 (0.0)	9 (100.0)		
**UICC**				0.083	
I	22 (11.3)	7 (31.8)	15 (68.2)		
II	67 (34.5)	31 (46.3)	36 (53.7)		
III	53 (27.3)	16 (30.2)	37 (69.8)		
IV	52 (26.8)	13 (25.0)	39 (75.0)		
**MSI status**				0.001 ***	**0.070**
MSS + MSI-L	112 (76.2)	43 (38.4)	69 (61.6)		
MSI-H	35 (23.8)	3 (8.6)	32 (91.4)		
**TP53**				0.887	
wt	21 (47.7)	4 (19.0)	17 (81.0)		
mut	23 (52.3)	4 (17.4)	19 (82.6)		
**K-Ras**				0.019 *	**0.005 ****
wt	105 (64.8)	39 (37.1)	66 (62.9)		
mut	57 (35.2)	11 (19.3)	46 (80.7)		
**B-Raf**				0.002 **	**0.004 ****
wt	139 (85.3)	49 (35.3)	90 (64.7)		
mut	24 (14.7)	1 (4.2)	23 (95.8)		
**PIK3CA**					
wt	16 (100.0)	4 (25.0)	12 (75.0)		
mut	0 (0.0)	0 (0.0)	0 (0.0)		
**C: PDTM**					
	**patients**	**model success**		**p-value**	
**n (%)**		**no**	**yes**	**univariate**	**Multivariate**
**age**				0.730	
<61	59 (19.8)	32 (54.2)	27 (45.8)		
61–70	73 (24.5)	36 (49.3)	37 (50.7)		
71–80	115 (38.6)	52 (45.2)	63 (54.8)		
>80	51 (17.1)	25 (49.0)	26 (51.0)		
**gender**				0.131	
male	178 (59.7)	93 (52.2)	85 (47.8)		
female	120 (40.3)	52 (43.3)	68 (56.7)		
**sample type**				0.150	
primary	254 (85.2)	128 (50.4)	126 (49.6)		
metastasis	44 (14.8)	17 (38.6)	27 (61.4)		
**localization**				0.011 *	
left colon	29 (9.7)	14 (48.3)	15 (51.7)		
rectum	35 (11.7)	20 (57.1)	15 (42.9)		
right colon	117 (39.3)	52 (44.4)	65 (55.6)		
sigmoid	59 (19.8)	39 (66.1)	20 (33.9)		
transverse	14 (4.7)	3 (21.4)	11 (78.6)		
metastasis	44 (14.8)	17 (38.6)	27 (61.4)		
**T**				0.061	
T0-2	44 (17.5)	28 (63.6)	16 (36.4)		
T3-4	208 (82.5)	100 (48.1)	108 (51.9)		
**N**				0.011 *	
N0	133 (52.8)	79 (59.4)	54 (40.6)		
N1	53 (21.0)	24 (45.3)	29 (54.7)		0.046 *
N2	66 (26.2)	25 (37.9)	41 (62.1)		0.110
**M**				0.008 **	**0.070**
M0	191 (75.8)	106 (55.5)	85 (44.5)		
M1	61 (24.2)	22 (36.1)	39 (63.9)		
**G**				0.006 **	
G1-2	173 (68.7)	98 (56.6)	75 (43.4)		
G3-4	79 (31.3)	30 (38.0)	49 (62.0)		
**R**				0.026 *	
R0	195 (77.4)	108 (55.4)	87 (44.6)		
R1	8 (3.2)	3 (37.5)	5 (62.5)		
R2	49 (19.4)	17 (34.7)	32 (65.3)		
**L**				0.688	
L0	183 (73.5)	94 (51.4)	89 (48.6)		
L1	66 (26.5)	32 (48.5)	34 (51.5)		
**V**				0.601	
V0	147 (59.0)	71 (48.3)	76 (51.7)		
V1	94 (37.8)	50 (53.2)	44 (46.8)		
V2	8 (3.2)	5 (62.5)	3 (37.5)		
**molecular type**				0.024 *	
spStd	93 (53.8)	44 (47.3)	49 (52.7)		
spMSI-H	33 (19.1)	9 (27.3)	24 (72.7)		
CIMP-H	37 (21.4)	11 (29.7)	26 (70.3)		
Lynch	10 (5.8)	1 (10.0)	9 (90.0)		
**UICC**				0.023*	
I	40 (14.8)	24 (60.0)	16 (40.0)		
II	93 (34.3)	57 (61.3)	36 (38.7)		
III	72 (26.6)	33 (45.8)	39 (54.2)		
IV	66 (24.4)	26 (39.4)	40 (60.6)		
**MSI status**				0.019 *	**0.070**
MSS + MSI-L	127 (75.1)	56 (44.1)	71 (55.9)		
MSI-H	42 (24.9)	10 (23.8)	32 (76.2)		
**TP53**				0.786	
wt	21 (45.7)	4 (19.0)	17 (81.0)		
mut	25 (54.9)	4 (16.0)	21 (84.0)		
**K-Ras**				0.014 *	**0.002 ****
wt	123 (65.1)	56 (45.5)	67 (54.5)		
mut	66 (34.9)	18 (27.3)	48 (72.7)		
**B-Raf**				0.022 *	**0.026 ***
wt	162 (84.8)	70 (43.2)	92 (56.8)		
mut	29 (15.2)	6 (20.7)	23 (79.3)		
**PIK3CA**					
wt	28 (100.0)	16 (57.1)	12 (42.9)		
mut	0 (0.0)	0 (0.0)	0 (0.0)		

Statistical analysis for correlations of model establishment success with clinicopathological for A: primary cell lines, B: PDX, and C: patient-derived tumor model (PDTM). Analysis were performed using a univariate (chi-squared-test and Fisher exact test) and multivariate approaches. The multivariate analysis was done including all significant parameters. * = *p* <0.05, ** = *p* < 0.01, *** = *p* < 0.001.

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
