# Peer review of "Integrated Biobanking and Tumor Model Establishment of Human Colorectal Carcinoma Provides Excellent Tools for Preclinical Research"

_cancers, 2019, doi:10.3390/cancers11101520_

Round 1
Reviewer 1 Report
The authors describe integrated biobanking and tumor model establishment for CRC.
A very nice and good initiative, well described.
I miss however the influence of the (current hot topic) of the microenvironment. Interesting to also have fibroblasts from the patients.
Author Response
We totally agree with the reviewer that the tumor microenvironment is one of the most interesting topics in the field. It was, however, not as hot more than a decade ago when we started our approach. As far as we are aware, fibroblasts have to be transformed in order to culture them for longer periods. Albeit we decided to go for EBV-transformation for patients’ B cells (to use them as antigen-presenting cells), we did not like the idea of using transformed-fibroblast co-cultures with autologous tumor cells. Similar to our line of argumentation with the tumor-organoids, we consider it possible to establish fibroblast cultures from vitally-frozen tumor pieces of the biobank when necessary and a suitable protocol is set-up. We added this point into the manuscript – and we additionally mentioned several other cell types recognized as important players in the tumor microenvironment: endothelial cells, tumor-associated macrophages, B cells and T cells.
It might be of interest to this reviewer that several of the cell lines tend to “happily” co-exist with fibroblasts at the beginning of the culturing process. This is typically a short and transient episode for primary patient-derived material – but it is regularly difficult to get rid of the mouse fibroblasts. And the latter are not exactly what we would aim at when going for tumor-associated fibroblast cultures.
Reviewer 2 Report
It is a interesting and relevant topic, however, some major revisions are necessary. First of all, there is a lack of a clear overview of the size of the final population what makes it very difficult to understand the steps the researchers have taken leading to the current results. It makes it also difficult to understand and interpret the results. As the title describes, the focus of the paper is on CRC. In line 85 the authors state that the CRC population consisted of 350 patients. However, at section 2.2 CRC cell cline was attempted in 388 tumors. Afterwards, in line 110 a number of 263 is mentioned but it’s not clear where this is coming from. Finally, in Table 1 the patient characteristics of 393 patients are described. This also includes patients that were not diagnosed with CRC. The authors should start section 2.1 with a clear description of the population they started with (n=393), the exclusion criteria plus the corresponding numbers and finally the population on which the findings are based. Afterwards, the actual results should be described and should be based on CRC patients only (e.g. section 2.2 and Table 1).
The authors are well aware that a biobank is a prerequisite for high-quality biomedical research. However, there is very little information available in the current paper on the actual biobank, the HROC collection. As a biobanker I want to know more of the procedures of the collection, processing, storage, and dissemination of biological samples plus the associated clinical data. How is this organized? For example, in section 2.1 the authors mention that they have collected outcome of disease. However, I miss detailed information regarding the process of data collection regarding outcome. Were there patients lost to follow-up? Was outcome data obtained by linking to national medical registries?
Minor revisions
Line 20: “…und lymphocytes” must be “…and lymphocytes” Line 54: please define “PCT” Line 68: please define “SOP” Line 85: the authors used the mean age plus the associated standard deviation. However, the population is most likely not distributed normally. It’s preferable to use the median plus range (minimum-maximum) instead
Author Response
First of all, there is a lack of a clear overview of the size of the final population what makes it very difficult to understand the steps the researchers have taken leading to the current results. It makes it also difficult to understand and interpret the results. As the title describes, the focus of the paper is on CRC. In line 85 the authors state that the CRC population consisted of 350 patients. However, at section 2.2 CRC cell cline was attempted in 388 tumors. Afterwards, in line 110 a number of 263 is mentioned but it’s not clear where this is coming from. Finally, in Table 1 the patient characteristics of 393 patients are described. This also includes patients that were not diagnosed with CRC. The authors should start section 2.1 with a clear description of the population they started with (n=393), the exclusion criteria plus the corresponding numbers and finally the population on which the findings are based. Afterwards, the actual results should be described and should be based on CRC patients only (e.g. section 2.2 and Table 1).
We removed the non-CRC tumors/patients from the manuscript and adapted the results part according to the reviewer’s suggestions. And we agree with the reviewer that this removal helped to make the results easier to understand for a reader.
The authors are well aware that a biobank is a prerequisite for high-quality biomedical research. However, there is very little information available in the current paper on the actual biobank, the HROC collection. As a biobanker I want to know more of the procedures of the collection, processing, storage, and dissemination of biological samples plus the associated clinical data. How is this organized? For example, in section 2.1 the authors mention that they have collected outcome of disease. However, I miss detailed information regarding the process of data collection regarding outcome. Were there patients lost to follow-up? Was outcome data obtained by linking to national medical registries?
This is a very helpful comment – we have to admit that we forgot to cite the original work describing the biobanking procedure itself. This has been modified accordingly.
The number of patients lost to follow-up had been listed in Table 1 – the “5-year follow-up rate”. It was 236 of the 350 CRC patients (67.4 %) – thus, 114 patients (32.6%) were lost to follow-up.
We also want to thank the reviewer for the hint towards linking to national medical registries. We approached the registry of the state Mecklenburg-Vorpommern – but due to the size of >100 patients, the response will last several weeks. Thus, we cannot address this point within the ten day-response time granted for sending in the revised version of the manuscript.
Line 20: “…und lymphocytes” must be “…and lymphocytes” Line 54: please define “PCT” Line 68: please define “SOP” Line 85: the authors used the mean age plus the associated standard deviation. However, the population is most likely not distributed normally. It’s preferable to use the median plus range (minimum-maximum) instead.
All minor points have been modified according to the reviewer’s suggestions.
Reviewer 3 Report
The authors give a good overview of a sound collection of CRC samples including cell lines and xenografts collected over ten years. The collection is described in large detail on how it was established and what factors could contribute to the success of creating the cell lines and xenografts. In addition it shows the collection covers all the histological subtypes.
The title however promises too much. The authors did not provide evidence or describe any research done showing this collection is an excellent tool for modern precision medicine. Although I am sure that using the collection for research in precision medicine it could be of great use, this part of the title should therefore be deleted. In the manuscript it is described that the collection took 10 years to set up, but in that time apparently no application was found in precision medicine. If so, it should be added to this manuscript.
Since the title refers to CRC, the non CRC samples leiomyosarcoma, lymphoma and mesenchymal tumor are not in place in the description and in the tables and should be deleted throughout the manuscript and tables
In table 2 A: primary cell line on top of the ages the complete row is missing that is needed to make clear that these group of numbers are ages and it also misses the p-value for this age group.
Some errors: Line 20 und should be and Line 181 ememplarily I think exemplary is meant.
Table 1 needs either in the legends explained that f=female and m=male or easier write it out in the heading of the table itself
Author Response
The title however promises too much. The authors did not provide evidence or describe any research done showing this collection is an excellent tool for modern precision medicine. Although I am sure that using the collection for research in precision medicine it could be of great use, this part of the title should therefore be deleted. In the manuscript it is described that the collection took 10 years to set up, but in that time apparently no application was found in precision medicine. If so, it should be added to this manuscript.
We have to partly agree with the reviewer that the title was a little exaggerating the data presented in the current manuscript and exchanged “modern precision medicine” with “preclinical research”.
In addition, we cited two recent results of cooperative work which very well can be considered as precision medicine applications.
Since the title refers to CRC, the non CRC samples leiomyosarcoma, lymphoma and mesenchymal tumor are not in place in the description and in the tables and should be deleted throughout the manuscript and tables.
We removed the non-CRC tumors/patients from the manuscript and adapted the results part according to the reviewer’s suggestions.
In table 2 A: primary cell line on top of the ages the complete row is missing that is needed to make clear that these group of numbers are ages and it also misses the p-value for this age group.
This has been corrected – many thanks for the attentive reading.
Some errors: Line 20 und should be and Line 181 ememplarily I think exemplary is meant.
Table 1 needs either in the legends explained that f=female and m=male or easier write it out in the heading of the table itself
The mistakes have been corrected and “female” and “male” have been written out in the heading of Table 1 as suggested.
Round 2
Reviewer 2 Report
My main comment on the previous version was that it was completely unclear what the size of the study population was. This makes it very difficult to understand and interpret the results. Although you made some changes it is still a puzzle to me.
Line 89: "...the CRC patient population (n=350)"
Line 93-96: The successfully biobanked material comprises 271 primary resected colorectal adenocarcinomas as well as 46 metastases from CRC. In addition, 9 adenomas and 5 neuroendocrine tumors (3 primaries and 2 metastases) were collected during this period.
This brings me at a total of 331. Does this mean that you did not collect tissue of 19 patients? Does this also mean that your final population consisted of 331 CRC patients, because you have biobanked their tissue successfully?
Line 115: Establishment of a primary (=patient derived) CRC cell line was attempted in 294/388 tumors.
The number of 388 has not been mentioned before and it is not clear to me where this comes from.
Line 169-170: In total, we could establish at least one patient-individual model (cell line and/or PDX) for 153/298 (51.3%) tumor samples.
The number of 298 has not been mentioned before and it is not clear to me where this comes from.
Finally, in Table 1 the patient characteristics of 387 patients are described. Why does this table still include patients that don’t belong to the final population that consisted of 350 CRC patients as you described in line 89?
Author Response
My main comment on the previous version was that it was completely unclear what the size of the study population was. This makes it very difficult to understand and interpret the results. Although you made some changes it is still a puzzle to me.
Line 89: "...the CRC patient population (n=350)"
On the one hand, we regret that we did not manage to completely solve these issues with the first revision. On the other hand, we have to thank this reviewer for his persistency, since the final version could be improved substantially in terms of clearness and logical order.
The patient population consisted of 382 patients diagnosed with a colorectal neoplasia (adenomas, primary and metastatic colorectal carcinomas). In all cases we collected at least blood samples and at times normal control tissue. From those initial 382 patients (regularly presenting repeatedly and/or with more than one neoplasia (adenoma and primary carcinoma, primary and metastatic carcinoma(s), etc.) we were able to collect and biobank 315 different neoplastic samples. In 107 cases (= 97 of the 382 patients) we were not able to collect neoplastic tissue samples of the patient operated on at that time point (not collected were 25 adenomas (from 24 patients) and 82 primary or metastatic carcinomas (72 primary & 10 metastatic carcinomas from 73 patients).
Line 93-96: The successfully biobanked material comprises 271 primary resected colorectal adenocarcinomas as well as 46 metastases from CRC. In addition, 9 adenomas and 5 neuroendocrine tumors (3 primaries and 2 metastases) were collected during this period.
This brings me at a total of 331. Does this mean that you did not collect tissue of 19 patients?
Does this also mean that your final population consisted of 331 CRC patients, because you have biobanked their tissue successfully?
We carefully re-checked and updated the numbers and the neuroendocrine samples were omitted. Moreover, we subdivided Supplementary Table 1 into two sheets. And please also refer to the answer above.
Line 115: Establishment of a primary (=patient derived) CRC cell line was attempted in 294/388 tumors.
The number of 388 has not been mentioned before and it is not clear to me where this comes from.
Line 169-170: In total, we could establish at least one patient-individual model (cell line and/or PDX) for 153/298 (51.3%) tumor samples.
The number of 298 has not been mentioned before and it is not clear to me where this comes from.
Several of the numbers counting the samples included into the different forms of model establishment are now explained better in the manuscript – and they can be picked in detail from the Supplementary Table 1, which we divided into two sheets to clarify this “sample” situation: sheet one shows all neoplastic samples (and corresponding patients) for which collection was successful and modelling could be attempted.
Sheet two lists the remaining attempts to collect neoplastic samples and the corresponding patients. Additionally, all adenomas can be found on sheet two, since so far no modelling attempts have been undertaken for adenomas in our lab. This may change in the future, as discussed in the manuscript, due to organoid cultures systems etc.
Finally, we listed also the neuroendocrine cases – we have the strong impression that some colleagues might be interested in this additional information. Of note, none of them were included into the statistical analysis of modelling success and clinical parameters.
Finally, in Table 1 the patient characteristics of 387 patients are described. Why does this table still include patients that don’t belong to the final population that consisted of 350 CRC patients as you described in line 89?
As outlined before we adapted numbers and added explanations on the “origin” of these numbers (detailed explanation please see above).
Round 3
Reviewer 2 Report
Dear Authors,
Thanks you for the changes you've made. It makes the structure of the paper much more logical. It is now also immediately clear what the results are based on.